

# Evaluation of resistance to powdery mildew and identification of resistance genes in wheat cultivars

Xianxin Wu[1,*], Qiang Bian[2,*], Yue Gao[1], Xinyu Ni[1], Yanqiu Sun[1], Yuanhu Xuan[1], Yuanyin Cao[1] and Tianya Li[1]

[1] College of Plant Protection, Shenyang Agricultural University Shenyang China
[2] National Pesticide Engineering Research Center, Nankai University Nanjing China
[*] These authors contributed equally to this work.

## ABSTRACT

Wheat powdery mildew, caused by the biotrophic fungus *Blumeria graminis* f. sp. *tritici* (*Bgt*), is a serious disease of wheat worldwide that can cause significant yield losses. Growing resistant cultivars is the most cost-effective and eco-soundly strategy to manage the disease. Therefore, a high breeding priority is to identify genes that can be readily used either singly or in combination for effective resistance to powdery mildew and also in combination with genes for resistance to other diseases. Yunnan Province, with complex and diverse ecological environments and climates, is one of the main wheat growing regions in China. This region provides initial inoculum for starting epidemics of wheat powdery mildew in the region and other regions and thus, plays a key role in the regional and large-scale epidemics of the disease throughout China. The objectives of this study were to evaluate seedling resistance of 69 main wheat cultivars to powdery mildew and to determine the presence of resistance genes *Pm3*, *Pm8*, *Pm13*, *Pm16*, and *Pm21* in these cultivars using gene specific DNA markers. Evaluation of 69 wheat cultivars with six *Bgt* isolates showed that only four cultivars were resistant to all tested isolates, indicating that the overall level of powdery mildew resistance of Yunnan wheat cultivars is inadequate. The molecular marker results showed that 27 cultivars likely have at least one of these genes. Six cultivars were found likely to have *Pm3*, 18 likely to have *Pm8*, 5 likely to have *Pm16*, and 3 likely to have *Pm21*. No cultivar was found to carry *Pm13*. The information on the presence of the *Pm* resistance genes in Yunnan wheat cultivars can be used in future wheat disease breeding programs. In particular, cultivars carrying *Pm21*, which is effective against all *Bgt* races in China, should be pyramided with other effective genes to developing new cultivars with durable resistance to powdery mildew.

# INTRODUCTION

Wheat (*Triticum aestivum* L.) is one of the most important food crops, which plays a very key role in the world food supply and food security, but its production is constantly challenged by various diseases (*Ma et al., 2014*; *Zhang et al., 2017*). Powdery mildew, caused by the biotrophic fungus *Blumeria graminis* f. sp. *tritici* (*Bgt*), is one of the most serious diseases

Corresponding author
Tianya Li, litianya11@syau.edu.cn

limiting wheat production in many regions of the world (*Zhang et al., 2016*). Breeding and growing resistant cultivars is generally considered to be the most economical, effective, and environmentally friendly method to control this disease (*Petersen et al., 2015*; *El-Shamy, Emara & Mohamed, 2016*). The first powdery mildew (*Pm*) resistance gene in wheat was found in wheat variety 'Thew' by Australian researcher Waterhouse in 1930 (*Zeller, 1973*). Since then, new powdery mildew resistance genes have been identified from common wheat and wheat relatives. In the meantime, the inheritance characteristics and chromosome locations of powdery mildew resistance genes were studied extensively (*Bhullar et al., 2010*; *Brunner et al., 2012*; *Hanusova et al., 1996*). To date, over 91 *Pm* resistance genes , mapped to 61 loci, have been characterized, and new genes are continually described in common wheat and relatives (*Hao et al., 2015*; *Li et al., 2017*; *Li et al., 2019*; *Tan et al., 2019*; *Zhang et al., 2017*). Many of these *Pm* genes have been used widely in wheat breeding programs (*Li et al., 2019*). Unfortunately, the *Pm* genes only confer resistance to specific *Bgt* races, and the race-specific nature is not ideal, since virulent mutants of *Bgt* can escape recognition of the resistance gene and making resistance genes ineffective (*Zhang et al., 2017*). For instance, the resistance gene *Pm8*, in a cluster with *Yr9*, *Lr26*, and *Sr31* for resistance to stripe rust, leaf rust, and stem rust, respectively, on 1BL/1RS was transferred into wheat cultivars from 'Petkus' rye in 1970s, and has a profound impact on wheat disease resistance breeding in the world (*Hurni et al., 2014*). Since then, a large number of wheat cultivars carrying 1BL/1RS have been released and widely grown in the world due to the resistance to multiple diseases. However, the overuse of the 1BL/1RS translocation in breeding and production has resulted in the rapid emergence of new pathotypes with the corresponding virulence genes, which have overcome the resistance genes, leading to serious epidemics of these diseases (*Mago et al., 2005*; *Pretorius et al., 2000*).

Recently, powdery mildew has become more significant with increased use of nitrogen fertilizer, changes in irrigation, and the increase of global average temperature (*Tang et al., 2017*). Therefore, cultivars become susceptible more quickly under the high disease pressure and more rapid changes of virulence in the pathogen population. Knowledge of the identity of race-specific resistance genes in wheat cultivars is a requirement to identify which resistance genes becoming ineffective. The use of molecular marker assisted selection breeding is a quick and easy approach to identify resistance genes. Molecular markers have been used to identify resistance genes against various diseases in wheat. Among various types of markers, simple sequence repeat (SSR) and single-nucleotide polymorphism (SNP) markers have recently been widely used in studying genes for resistance to powdery mildew (*Keller et al., 1999*; *Liu et al., 2002*; *Wu et al., 2019*).

Yunnan Province, located in the southwest of China, has complex and diverse ecological environments and climates. In this region, wheat powdery mildew is very serious and epidemic occurs every year. Because of the disease-favorable environments, Yunnan provides initial inoculum for wheat powdery mildew, stem rust, leaf rust, and stripe rust, playing a key role in the regional spread and large-scale epidemic of the diseases in China (*Li et al., 2012*; *Li et al., 2016*). Therefore, assessment of the resistance level of the main production cultivars to these diseases as well as identification of resistance genes in the cultivars can provide a theoretical basis for diseases management by rationally deploying

**Table 1  Pedigrees of 69 wheat cultivars used in this study.**

| Cultivar/line | Pedigree | Cultivar/line | Pedigree |
| --- | --- | --- | --- |
| 017-10 | Bolsena-1CH/[SieteCerros/XBVT223//AWX011.G.48.2/XBVT221] | Mianyang 19 | Selected from Fan 6/70-5858 with systemic selection |
| 02D2-282 | Yunmai 39/Yunmai 42 | Mentana | Reiti /Wilhelmina// Akagomughi |
| 06D6-6 | Yumai 39/992-3 | Mian1971-98 | 96-18-6/92R178 |
| 91E001 | Selected from Mexico wheat | Yunmai 43 | Unkonwn |
| Chumai 12 | Selected from the observation materials of 02d-195 with systemic selection | Mianyang 20 | Selected from 70-5858/ Fan 6 with systemic selection |
| De 4-8 | Unknown | Nanyuan 1 | Mentana /Yuannong 1 |
| De 05-81 | 9213-194/9213-4074 | R101 | AGA/HORK"S" |
| De 08-3 | Unkownn | R57 | Selected from Mexico wheat |
| Wenmai 12 | Selected from the hybrid advanced lines of '0581-1' with systemic selection | Wenmai 11 | Selected from the hybrid advanced lines of '0581-39' with systemic selection |
| Demai 3 | Longchun 2/Mocha | Demai 4 | 782-88/Zhongyin 1022 |
| Demai 5 | Mianyang 11/Yun 80-1 | Yimai 1 | Unknown |
| Yimai lines 2003-13 | Chuanmai 24/96-16 | Yimai 10 | Selected from Yimai 1 with systemic selection |
| Demai 7 | Yunzhi 437/892- 17 | E33 | Selected from Mexico wheat |
| Feng 05-394 | Selected from Fengmai 31's with single systemic selection | Yimai lines 2003-27 | 96-23/96-14 |
| Feng 1124 | E33/58769-6 | Yixi 2003-64 | 96-23/96-14 |
| Fengmai 13 | Unknown | Yumai 1 | Xingaibai/Germanic dunmai |
| Fengmai 24 | Moba 65/Precocious Ajin/Mosha F $_6$/750025-12/ Mexico advanced lines 965 | Fengmai 36 | Selected from Fengmai 31's with single systemic selection |
| Yunmai 29 | Zhushi Wheat/Fuli Wheat | Yumai 3 | 82-1/8334 |
| Fengmai 34 | 9034M3-2-2/YV91-1167 | Yunmai 101 | 963-8224/98042-7 |
| Fengmai 35 | Fengmai 24//806-14-2-15/85-7421 F1 | Yunmai 11-12 | NG8319//SHA4/LTRA |
| Fengmai 37 | 9034M8-17/Fengmai 24 | Yunmai 39 | Zhushi Wheat/Fuli Wheat/Fticher's |
| Fengmai 38 | Momai Lines 91E001/Advanced lines 8941 | Yunmai 42 | Kangxiu 782/Yunmai 29//YKLO-PAM S |
| Fengmai 39 | Selected from Fengmai 31's with single systemic selection | Yunxuan 3 | SW8488*2/4/SIN/TRAP#1/3/KAUZ*2/TRAP//KAUZ |
| Jingmai 11 | Kavkaz 78-385/Mo 980// Multi-parent mixed pollen | Yunmai 47 | 852-18/852-181//86-4437-75/3/822-852/785//842-929/4/Gangu 436/5/923-3763 |
| Jing 0202 | Jingmia 10/96 Feng 1 | Yunmai 48 | 48 99213/92B-4074 |
| Jing 06-4 | 849M$_2$-11-23/7730-1-149 | Yunmai 51 | 91B-831/92B-84 |

**Table 1** (*continued*)

| Cultivar/line | Pedigree | Cultivar/line | Pedigree |
|---|---|---|---|
| Fengyin 03-2 | Unknown | Yunmai 52 | 92R149/963-11185 |
| Jingmai 12 | 7901/792364//9118 | Yunmai 53 | 96B-254/96B-6 |
| Jingmai 14 | Sumai 3/Qing 30// 8619-10 | Yunmai 54 | Yunmai 39/S-792 |
| Kun 022-222-1 | Selected from the advanced lines 022-222 with systemic selection | Yunmai 56 | Advanced lines 932-625/A dvanced lines 822-16-7-3 |
| Kunmai 4 | Selected from the kunmai 2 with systemic selection | Yunmai 57 | Screening wheat materials from CIMMYT |
| Kunmai 5 | 992-17/Huelauen | Yunza 5 | 01Y1-1069/K78S |
| Liangmai 4 | N1491/N1071 | Yunza 6 | K78S/01Y1-608 |
| Linmai 15 | A122/(87-5/E232) | Yunza 7 | K78S/02Y1-101 |
| Linmai 6 | 86 Jian 22/84-346 | | |

cultivars with various resistance genes in different areas. Resistance to stripe rust in Yunnan wheat cultivars has been studied by *Li (2013)*. In our previous study, resistance to stem rust in main wheat cultivars of the region was also studied (*Li et al., 2016*; *Xu et al., 2017*). In recent years, the epidemic level of powdery mildew has been increasing in Yunnan (*Tang et al., 2017*). Therefore, this study was carried out to determine the level of seedling resistance to powdery mildew and to identify *Pm* genes in wheat cultivars using molecular markers. This information will be useful for developing wheat cultivars with durable resistance to powdery mildew.

## MATERIAL AND METHODS

### Wheat cultivars and *Pm* resistance lines

A total of 69 wheat cultivars and breeding lines used in the present study included main cultivars grown in Yunnan province and genetic stocks used in breeding programs, and seeds were provided by Pro. Mingju Li, Institute of Agricultural Environment and Resources, Yunnan Academy of Agricultural Sciences. The pedigrees of the cultivars are listed in Table 1. A set of 37 wheat lines carrying known powdery mildew resistance genes (Table 2) were also used in the present study as *Pm* gene references, and seeds were provided by Prof. Yilin Zhou, State Key Laboratory for Biology of Plant Diseases and Insect Pests, Institute of Plant Protection, Chinese Academy of Agricultural Sciences.

### Isolates of *B. graminis* f. sp. *tritici*

Six isolates of *B. graminis* f. sp. *tritici* with different virulence patterns were used to evaluate resistance in the wheat cultivars and breeding lines. These isolates were selected from the collection of the Plant Immunity Institute, Shenyang Agricultural University, China. Their virulence/avirulence patterns to the 37 wheat differentials carrying known *Pm* genes are shown in Table 2.

### Evaluation of seedling resistance

The 69 wheat cultivars and breeding lines were evaluated in the seedling stage for resistance to powdery mildew using the six *Bgt* isolates in the greenhouse at the College of Plant Protection, Shenyang Agricultural University, using the method described in a previous

**Table 2** Infection types of 37 wheat genotypes with known *Pm* genes to tested isolates of *Blumeria graminis* f. sp. *tritici*.

| Cultivars (line) | *Pm* gene | Infection types to *B. graminis* f. sp. *tritici* isolates[a]. | | | | | |
|---|---|---|---|---|---|---|---|
| | | 09558-1 | W1 | W12 | L14 | T7 | H1-5-1 |
| Aminster/8cc | *Pm1* | 4 | 0 | 3 | 4 | 4 | 3 |
| Ulka/8cc | *Pm2* | 4 | 3 | 4 | 0 | 3 | 4 |
| Asosan/8cc | *Pm3a* | 4 | 0 | 3 | 4 | 4 | 3 |
| Chul/8cc | *Pm3b* | 4 | 0 | 3 | 3 | 4 | 0 |
| Sonora/8cc | *Pm3c* | 4 | 3 | 4 | 4 | 4 | 3 |
| Kolibri | *Pm3d* | 4 | 0 | 1 | 1 | 4 | 4 |
| W150 | *Pm3e* | 4 | 0 | 4 | 3 | 4 | 4 |
| Mich.Amber/8cc | *Pm3f* | 4 | 4 | 4 | 2 | 4 | 4 |
| Whapli/8cc | *Pm4a* | 4 | 1 | 3 | 1 | 4 | 4 |
| Armada | *Pm4b* | 4 | 3 | 4 | 3 | 4 | 4 |
| Hope/8cc | *Pm5* | 4 | 1 | 4 | 1 | 4 | 3 |
| Coker983 | *Pm5+6* | 4 | 2 | 0 | 4 | 0 | 3 |
| Tingalen | *Pm6* | 4 | 3 | 1 | 2 | 3 | 2 |
| Coker747 | *Pm6* | 4 | 3 | 3 | 3 | 4 | 3 |
| CI14189 | *Pm7* | 4 | 0; | 1 | 3 | 4 | 4 |
| Kavkaz | *Pm8* | 4 | 3 | 3 | 2 | 0 | 0; |
| Kenguia | *Pm4+8* | 4 | 3 | 4 | 3 | 4 | 3 |
| Normandie | *Pm1+2+9* | 4 | 3 | 1 | 3 | 4 | 4 |
| R4A | *Pm13* | 0; | 1 | 0 | 0 | 0; | 1 |
| Brigand | *Pm16* | 3 | 0 | 3 | 1 | 3 | 4 |
| MIN | *Pm18* | 0 | 1 | 2 | 0 | 0 | 1 |
| KS93WGRC28 | *Pm20* | 4 | 3 | 3 | 3 | 4 | 3 |
| Yangmai 5/sub.6v | *Pm21* | 0 | 0 | 0 | 0 | 0 | 1 |
| Virest | *Pm22* | 0 | 0 | 4 | 1 | 0 | 2 |
| line81-7241 | *Pm23* | 4 | 0; | 3 | 1 | 4 | 3 |
| Chiyacao | *Pm24* | 4 | 3 | 3 | 0 | 4 | 3 |
| 5P27 | *Pm30* | 4 | 3 | 4 | 1 | 4 | 4 |
| Mission | *Pm4b+mli* | 4 | 3 | 4 | 0 | 4 | 4 |
| Maris Dire | *Pm2+mld* | 3 | 1 | 4 | 0 | 0 | 3 |
| Xiaobaidongmai | *PmXBD* | 4 | 1 | 4 | 0 | 1 | 1 |
| Baimian 3 | *Pm4+8* | 3 | 1 | 3 | 0 | 4 | 4 |
| CI12632 | *Pm4+8* | 4 | 1 | 4 | 0 | 4 | 3 |
| Maris Huntsman | *Pm2 + 6 +* | 3 | 4 | 3 | 4 | 1 | 3 |
| Era | – | 4 | 3 | 4 | 0 | 0 | 2 |
| Amigo | *Pm17* | 4 | 1 | 1 | 3 | 4 | 4 |
| XX186 | *Pm19* | 4 | 3 | 4 | 3 | 4 | 4 |
| Funo | - | 4 | 4 | 4 | 3 | 4 | 4 |

**Notes.**

[a]Infection types: 0 = no visible symptoms; 0; = hypersensitive necrotic flecks; 1 = minute colonies with few conidia; 2 = colonies with moderately developed hyphae, but few conidia; 3 = colonies with well-developed hyphae and abundant conidia, but colonies not joined together; and 4 = colonies with welldeveloped hyphae and abundant conidia, and colonies mostly joined together.

Table 3  Molecular markers linked to resistance genes *Pm3*, *Pm8*, *Pm13*, *Pm6*, and *Pm21* with their forward and backward primers.

| Tagged *Pm* gene | Primer | Fragment size (bp) | Primer sequence (5′–3′) |
|---|---|---|---|
| *Pm3* | Pm3a | 624 | GGA GTC TCT TCG CAT AGA CAG CTT CTA AGA TCA AGG AT |
| *Pm8* | IAG95 | 1050 | AGCAACCAAACACACCCATC ATACTACGAACACACACCCC |
| *Pm13* | UTV14F/R | 517 | CGCCAGCCAATTATCTCCATGA AGCCATGCGCGGTGTCATGTGAA |
| *Pm16* | Xgwm159 | 201 | GGGCCAACACTGGAACAC GCAGAAGCTTGTTGGTAGGC |
| *Pm21* | Scar1265 | 1265 | CACTCTCCTCCACTAACAGAGG GTTTGTTCACGTTGAATGAATC |

study (*Xiang et al., 1994*). About 10 seeds of each cultivar were sown in a pot of 12 cm in diameter. Highly susceptible cultivar Chancellor was used as a control for evaluating uniformity of inoculation. Plants in different trays were inoculated with the six *Bgt* isolates separately, and each tray was covered with a glass shroud to avoid cross infection of different strains after inoculation, when the primary leaves were fully expanded (about 10 days after planting). When the susceptible cultivar Chancellor was heavily infected, about 10 days after inoculation, infection types (ITs) were recorded. A 0-to-4 ITs scale was used for recording the host response to infection (*Si et al., 1987*), where 0= no visible symptoms; 0; = hypersensitive necrotic flecks; 1 = minute colonies with few conidia produced; 2 = colonies with moderately developed hyphae, but few conidia; 3 = colonies with well-developed hyphae and abundant conidia, but colonies not joined together; and 4 = colonies with well-developed hyphae and abundant conidia, and colonies mostly joined together. ITs 0–2 were considered as 'R' (resistant), and ITs 3–4 as 'S' (susceptible).

### DNA extraction and PCR amplification

Genomic DNA was extracted from 100 mg young leaves of seven-day old seedlings from each cultivar, using a DNA extraction kit (Sangon Biotech, Shanghai, CHINA). *Pm*-gene specific primers were synthesized by Shanghai Biotech Biotech Co., Ltd, China (Table 3). Polymerase chain reactions (PCR) were carried out using a S1000$^{TM}$ Thermal Cycler in 25 μL volume, including 2 μL of 50 ng μL$^{-1}$ DNA, 1 μL of 10 μmol L$^{-1}$ of each primer, 2.5 μL of 10 × buffer (Mg$^{2+}$), 0.2 μL of 5 U μL$^{-1}$ *Taq* polymerase, and 0.5 μL of 10 mmol · L$^{-1}$ deoxyribonucleoside triphosphates. The PCR procedure was as follows: 94 °C for 5 min, 30 cycles of 94 °C for 45 s, 60 °C for 45 s, and 72 °C for 1 min, followed by the final extension at 72 °C for 8 min. PCR products were separated on 1.5–2% agarose.

## RESULTS

### Wheat seedling resistance to *B. graminis* f. sp. *tritici*

The powdery mildew infection types of 69 main wheat cultivars and breeding lines to all tested isolates were presented in Table 4. Five wheat cultivars, De 4-8, Kunmai 4, Yixi
**Table 4  Percentages of susceptible and resistant wheat cultivars to six isolates of *Blumeria graminis* f. sp. *tritici*.**

| Isolates | Susceptible | | Resistance | |
|---|---|---|---|---|
| | Number of cultivars | Percentage (%) | Number of cultivars | Percentage (%) |
| 09558-1 | 61 | 88.4 | 8 | 11.6 |
| W1 | 46 | 66.7 | 23 | 33.3 |
| W12 | 54 | 78.3 | 15 | 21.7 |
| L14 | 54 | 78.3 | 15 | 21.7 |
| T7 | 56 | 81.2 | 13 | 18.8 |
| H1-5-1 | 61 | 88.4 | 8 | 11.6 |
| All tested isolates | 65 | 94.2 | 4 | 5.8 |

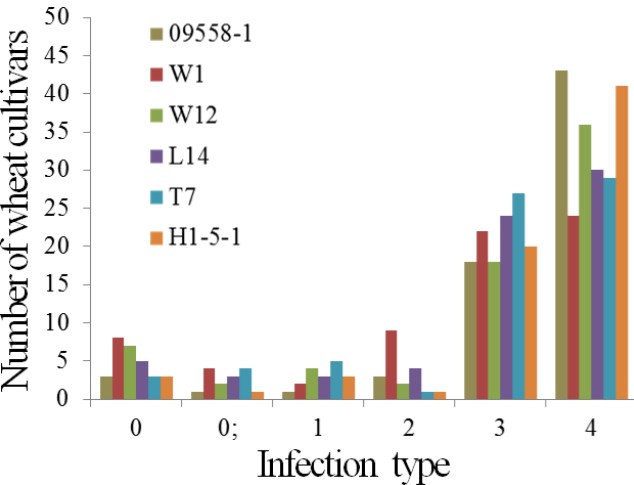

**Figure 1  Number of wheat cultivars and breeding lines showing different infections in seedlings when tested with six isolates of *Blumeria graminis* f. sp. *tritici*.**

2003-64, Yimai lines 2003-13, and Yimai lines 2003-27 were resistant (ITs 0-2) to all tested isolates at seedling stage, accounting for only 5.8% of the tested cultivars and breeding lines. The remaining 64 wheat cultivars were susceptible to one or more tested isolates (Fig. 1, Table 4).

## Molecular identification of *Pm3*

*Pm3* is a single, dominant locus on the short arm of wheat chromosome 1A and contains more alleles than any other identified *Pm* loci (*Tommasini et al., 2006*). Seven specific markers for the *Pm3* alleles (*Pm3a - Pm3g*) based on nucleotide polymorphisms of coding and adjacent noncoding regions were used to identify *Pm3* alleles in the wheat cultivars. The specific fragment of 624-bp for *Pm3a* was amplified in the positive control Asonsan/8cc that is known to carry *Pm3a* and six cultivars (Fengmai 35, Jing 0202, Liangmai 4, Wenmai 11, Yumai 3, and Yunmai 51), indicating that these cultivars are most likely to carry *Pm3a* (Fig. 2A, Table 5).
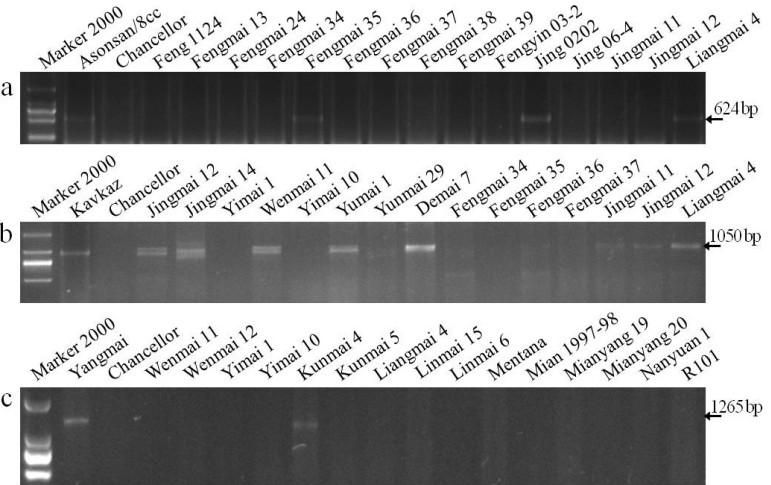

**Figure 2** **Electrophoretograms of primers for different *Pm* genes.** The corresponding primers for *Pm3a* (A), *Pm8* (B), and *Pm21* (C) were used to screen Yunnan wheat cultivars and breeding lines.

## Molecular identification of *Pm8*

The best known and widely deployed *Pm8* is located on a 1BL.1RS translocation in hexaploid wheat. It was originally derived from the introgression of the 1RS rye chromosome from rye cultivar 'Petkus'. A sequence-tagged site (STS) marker, IAG95, was developed to identify *Pm8* (*Mohler et al., 2001*). Wheat cultivars with the chromatin from 'Petkus' carry the resistance allele amplified as 1,050 bp fragment by the IAG95 primer pair. In this study, the positive control 'Kavkaz' (*Pm8*) and 18 wheat cultivars had the 1,050 bp fragment, indicating that these cultivars are most likely to carry *Pm8* (Fig. 2B).

## Molecular identification of *Pm13*

*Cenci et al. (1999)* amplified a 517-bp specific fragment in cultivars containing the *Pm13* genes by using STS UTV14F/R primers. A 517-bp fragment was amplified in the wheat line R4A (*Pm13*) as a positive control, and no fragment was amplified in the negative control Chancellor as expected. However, no specific fragment was amplified in all tested wheat cultivars and breeding lines, indicating that none of these wheat varieties (lines) are most likely to carry *Pm13*.

## Molecular identification of *Pm21*

*Liu et al. (1999)* detected a 1265-bp fragment in Yangmai 5/Sub−6 V (*Pm21*) by amplification using the STS 1265 marker of *Pm21*. In our study, this specific fragment was amplified in the *Pm21* positive control, but not in the susceptible control Chancellor. This gene was also detected in cultivars Kunmai 4, Yixi 2003-64, and De 4-8, but not in the rest of the cultivars and breeding lines, indicating that these three Yunnan cultivars are most likely to contain *Pm21* (Fig. 2C).

Wu et al. (2021), *PeerJ*, DOI 10.7717/peerj.10425

**Table 5  Seedling infection types to *Blumeria graminis* f. sp. *tritici* and resistance genes detected with molecular markers.**

| Cultivars/lines | Infection types[a] | | | | | | Resistance gene | | | | |
|---|---|---|---|---|---|---|---|---|---|---|---|
| | 09558-1 | W1 | W12 | L14 | T7 | H1-5-1 | *Pm3* | *Pm8* | *Pm13* | *Pm16* | *Pm21* |
| 017-10 | 4 | 3 | 4 | 3 | 3 | 4 | − | − | − | − | − |
| 02D2-282 | 4 | 4 | 4 | 4 | 4 | 4 | − | − | − | − | − |
| 06D6-6 | 4 | 4 | 4 | 4 | 4 | 3 | − | − | − | − | − |
| 91E001 | 4 | 3 | 3 | 4 | 3 | 4 | − | − | − | − | − |
| Chumai 12 | 3 | 4 | 3 | 4 | 4 | 4 | − | − | − | − | − |
| De 4-8 | 0 | 0 | 0; | 0 | 0; | 0 | − | − | − | − | + |
| De 05-81 | 3 | 4 | 4 | 3 | 4 | 3 | − | + | − | − | − |
| De 08-3 | 2 | 0 | 0 | 1 | 0; | 0 | − | − | − | − | − |
| Demai 3 | 4 | 3 | 4 | 3 | 3 | 3 | − | − | − | − | − |
| Demai 4 | 3 | 3 | 4 | 4 | 4 | 4 | − | + | − | − | − |
| Demai 5 | 4 | 4 | 4 | 3 | 3 | 4 | − | − | − | − | − |
| Demai 7 | 4 | 4 | 4 | 4 | 4 | 4 | − | + | − | − | − |
| E33 | 4 | 3 | 4 | 4 | 3 | 3 | − | + | − | − | − |
| Feng 05-394 | 4 | 3 | 4 | 4 | 3 | 4 | − | − | − | − | − |
| Feng 1124 | 3 | 2 | 3 | 0 | 3 | 3 | − | − | − | + | − |
| Fengmai 13 | 4 | 3 | 4 | 3 | 4 | 4 | − | − | − | − | − |
| Fengmai 24 | 4 | 4 | 4 | 4 | 4 | 4 | − | − | − | − | − |
| Fengmai 34 | 3 | 3 | 4 | 4 | 4 | 4 | − | − | − | − | − |
| Fengmai 35 | 4 | 2 | 0 | 3 | 3 | 3 | + | − | − | − | − |
| Fengmai 36 | 4 | 3 | 4 | 4 | 3 | 3 | − | − | − | − | − |
| Fengmai 37 | 4 | 0 | 2 | 3 | 0 | 3 | − | − | − | − | − |
| Fengmai 38 | 4 | 0; | 0 | 4 | 3 | 4 | − | − | − | − | − |
| Fengmai 39 | 4 | 3 | 4 | 2 | 4 | 3 | − | − | − | − | − |
| Fengyin 03-2 | 4 | 2 | 4 | 2 | 3 | 4 | − | − | − | + | − |
| Jing 0202 | 3 | 2 | 3 | 4 | 0 | 4 | + | + | − | − | − |
| Jing 06-4 | 3 | 3 | 4 | 3 | 4 | 4 | − | + | − | − | − |
| Jingmai 11 | 4 | 0 | 0; | 3 | 4 | 3 | − | + | − | − | − |
| Jingmai 12 | 4 | 3 | 4 | 4 | 3 | 4 | − | + | − | − | − |
| Jingmai 14 | 4 | 4 | 3 | 4 | 4 | 4 | − | + | − | − | − |
| Kun 022 − 222 − 1 | 4 | 4 | 3 | 3 | 4 | 4 | − | − | − | − | − |

Wu et al. (2021), *PeerJ*, DOI 10.7717/peerj.10425

**Table 5** (*continued*)

| Cultivars/ lines | Infection types[a] | | | | | | Resistance gene | | | | |
|---|---|---|---|---|---|---|---|---|---|---|---|
| | 09558-1 | W1 | W12 | L14 | T7 | H1-5-1 | *Pm3* | *Pm8* | *Pm13* | *Pm16* | *Pm21* |
| Kunmai 4 | 0 | 0; | 0 | 1 | 0 | 1 | – | – | – | – | + |
| Kunmai 5 | 4 | 3 | 0 | 0; | 4 | 4 | – | + | – | – | – |
| Liangmai 4 | 4 | 2 | 4 | 1 | 3 | 3 | + | + | – | – | – |
| Linmai 15 | 3 | 4 | 4 | 4 | 3 | 4 | – | + | – | – | – |
| Linmai 6 | 4 | 4 | 3 | 4 | 4 | 4 | – | – | – | – | – |
| Mentana | 3 | 3 | 4 | 3 | 3 | 3 | – | – | – | – | – |
| Mian 1971-98 | 4 | 0; | 3 | 4 | 4 | 3 | – | – | – | – | – |
| Mianyang 19 | 3 | 3 | 4 | 3 | 4 | 4 | – | – | – | – | – |
| Mianyang 20 | 4 | 4 | 3 | 4 | 4 | 4 | – | – | – | – | – |
| Nanyuan 1 | 4 | 3 | 3 | 4 | 3 | 4 | – | – | – | – | – |
| R101 | 3 | 4 | 3 | 3 | 3 | 3 | – | – | – | – | – |
| R57 | 4 | 0 | 4 | 3 | 2 | 4 | – | + | – | – | – |
| Wenmai 11 | 4 | 0; | 4 | 3 | 1 | 4 | + | + | – | – | – |
| Wenmai 12 | 4 | 3 | 1 | 2 | 4 | 3 | – | + | – | – | – |
| Yimai 1 | 4 | 4 | 4 | 4 | 4 | 4 | – | – | – | – | – |
| Yimai 10 | 3 | 4 | 4 | 4 | 4 | 4 | – | – | – | – | – |
| Yimai lines 2003-13 | 1 | 0 | 1 | 0; | 1 | 1 | – | – | – | – | – |
| Yimai lines 2003-27 | 0 | 1 | 1 | 0; | 1 | 0; | – | – | – | – | – |
| Yixi 2003-64 | 0; | 0 | 1 | 0; | 1 | 0 | – | – | – | – | + |
| Yumai 1 | 4 | 4 | 4 | 4 | 4 | 4 | – | + | – | – | – |
| Yumai 3 | 3 | 2 | 4 | 3 | 3 | 4 | + | + | – | – | – |
| Yunmai 101 | 3 | 4 | 3 | 3 | 4 | 4 | – | + | – | – | – |
| Yunmai 11-12 | 4 | 3 | 4 | 4 | 3 | 4 | – | – | – | – | – |
| Yunmai 29 | 4 | 3 | 4 | 3 | 4 | 4 | – | – | – | – | – |
| Yunmai 39 | 3 | 3 | 2 | 3 | 4 | 4 | – | – | – | + | – |
| Yunmai 42 | 3 | 3 | 4 | 3 | 3 | 4 | – | – | – | + | – |
| Yunmai 43 | 4 | 4 | 3 | 3 | 4 | 1 | – | – | – | – | – |
| Yunmai 47 | 3 | 4 | 4 | 4 | 3 | 3 | – | – | – | – | – |
| Yunmai 48 | 2 | 4 | 3 | 0 | 4 | 4 | – | – | – | – | – |

Wu et al. (2021), *PeerJ*, DOI 10.7717/peerj.10425

**Table 5** (*continued*)

| Cultivars/lines | Infection types[a] | | | | | | Resistance gene | | | | |
|---|---|---|---|---|---|---|---|---|---|---|---|
| | 09558-1 | W1 | W12 | L14 | T7 | H1-5-1 | *Pm3* | *Pm8* | *Pm13* | *Pm16* | *Pm21* |
| Yunmai 51 | 4 | 0; | 4 | 3 | 3 | 3 | + | − | − | − | − |
| Yunmai 52 | 4 | 4 | 3 | 4 | 3 | 4 | − | − | − | − | − |
| Yunmai 53 | 3 | 2 | 3 | 0 | 0; | 1 | − | − | − | − | − |
| Yunmai 54 | 4 | 3 | 4 | 4 | 3 | 4 | − | − | − | − | − |
| Yunmai 56 | 3 | 0 | 0 | 4 | 1 | 3 | − | − | − | − | − |
| Yunmai 57 | 4 | 4 | 0 | 3 | 0; | 4 | − | − | − | − | − |
| Yunxuan 3 | 4 | 3 | 3 | 4 | 4 | 4 | − | − | − | − | − |
| Yunza 5 | 4 | 4 | 3 | 3 | 3 | 3 | − | − | − | − | − |
| Yunza 6 | 4 | 4 | 4 | 3 | 4 | 3 | − | − | − | − | − |
| Yunza 7 | 4 | 3 | 4 | 4 | 3 | 4 | − | − | − | − | − |
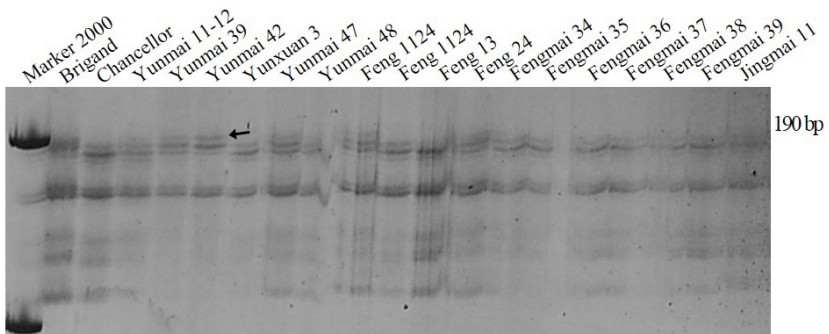

190 bp

**Figure 3** **Amplification result for parts of wheat varieties amplified with premier Xgwm159 of *Pm16*.**

## Molecular identification of *Pm16*

SSR marker *Xgwm159* for *Pm16* developed by *Chen et al. (2005)* was used to test wheat cultivars and breeding lines. PAGE (polyacrylamide gel electrophoresis) showed that the fragment amplified by the varieties containing *Pm16* was about 201-bp, while the fragment amplified by the varieties without *Pm16* was about 190-bp. Cultivars Yunmai 39, Yunmai 42, Yunmai 47, Feng 1124 and Fengyin 03-2 had the same fragment as the positive control of the *Pm16* single-gene line indicating that these five cultivars are most likely to contain *Pm16* (Fig. 3).

## DISCUSSION

Since the 1970s, wheat powdery mildew has been prevalent all over the world, causing different degrees of economic losses every year. Developing powdery mildew resistant cultivars has always been an important breeding goal of wheat. Powdery mildew has been well controlled, and the losses caused by the disease have been reduced in different stages in history. However, in recent years, wheat powdery mildew has been increasing as a result of losses of varietal resistance caused by the high heterogeneity and frequent virulence changes in the pathogen population. Many wheat cultivars planted at present show a trend of high susceptibility to powdery mildew. Therefore, it is urgent to improve resistance to powdery mildew. In the present study, the resistance level of wheat cultivars in Yunnan province to six *Bgt* isolates was evaluated. The results showed that most of tested wheat cultivars were highly susceptible to *Bgt*, indicating that effective genes are lack in Yunnan cultivars. In addition, currently effective genes (*Pm13*, *Pm16*, and *Pm21*) to *Bgt* in Chinese wheat cultivars are in very low frequencies as only four cultivars contain *Pm16*, and 3 cultivars contain *Pm21*, while no cultivars contain *Pm13*. Therefore, in order to improve the resistance level of wheat cultivars to powdery mildew, it is necessary to pyramid some other effective genes into new cultivars.

The *Pm3* locus was one of the first described loci for resistance to powdery mildew (*Briggle & Sears, 1966*). Some of the resistance alleles have been widely used in wheat breeding programs in the many countries including China, and some of the resistance alleles have remained effective (*Bougot et al., 2002*; *Wu et al., 2019*). *Liu et al. (2019)* reported about

95.1% wheat cultivars from Heilongjiang Province carrying *Pm3*. In the present study, we identified the gene in only six cultivars grown in Yunnan (Fengmai 35, Jing 0202, Liangmai 4, Wenmai 11, Yumai 3, and Yunmai 51).

The gene *Pm8* on 1BL/1RS was transferred into many bread wheat cultivars from 'Petkus' rye (*Graybosch, 2001*). The 1BL/1RS translocation has been playing an important role in wheat disease resistance breeding in the world, because this locus is closely linked to disease resistance genes, including *Sr31*, *Lr26*, and *Yr9* for resistance to stem rust, leaf rust, and stripe rust, respectively. It is reported that more 50% of the wheat cultivars grown in total wheat planting areas in China carry this translocation (*Li et al., 2011*). Our results showed that eighteen wheat cultivars contain *Pm8*, accounting for 26.1% of the tested cultivars from Yunnan. Conversely, pedigree tracking indicated that resistant stocks carrying *Pm 8*, such as 'Kavkaz' and 'Lovrin' lines, were widely used in wheat breeding in Yunnan Province, suggesting the origin of resistance genes in these wheat cultivars (*Li et al., 2016*). Our results were consistent with previous reports. For example, *Liu et al. (2019)* found that the frequency of 1BL/1R translocation in Huang-Huai wheat region was as high as 59%. In addition, no virulent races of *P. graminis* f. sp. *tritici* to resistance gene *Sr31* has been found in China. Therefore, this gene will still have an impact on wheat breeding for disease resistance, although the resistance to *Bgt* has been lost in China. Thus, *Pm8* should be used in combination with other genes for effective resistance to *Bgt* in wheat breeding programs to maintain the long-term resistance of cultivars.

Gene *Pm13* originated from *Aegilops longissima* and was located on the 6VS of the translocation chromosome T6AL/6VS of wheat/ *Aegilops longissima* translocation. It is one of the effective resistance genes to powdery mildew in the world including China. *Cenci et al. (1999)* was first developed the STS linkage marker of *Pm13*, which is widely used in marker assisted selection breeding. In our previous study, *Pm13* was found to be effective in northeastern China (*Wu et al., 2019*). However, *Pm13* was not found in any of the Yunnan wheat cultivars tested in the present study. Similarly, *Li et al. (2009)* and *Liu et al. (2010)* did not detected *Pm13* in any of the cultivars, including 50 and 101 cultivars from different regions of China. Their results, together with our study, indicate that *Pm13* is absent in Chinese wheat cultivars, and this effective gene should be used in breeding programs.

Gene *Pm16* was the first wheat powdery mildew resistance genes transferred from *Triticum dicoccoides* Korn into *Triticum aestivum* L. and was first reported in chromosome 4A at the earliest (*Reader & Miller, 1991*). However, subsequent studies did not show a consistent chromosomal location. *Wang (2004)* reported the gene on 5DS, while *Chen et al. (2005)* reported it on 5BS. Therefore, multiple markers have been reported for *Pm16*. However these markers may not be specific. In the present study, the SSR marker reported by *Chen et al. (2005)* was used, and four cultivars, Yunmai 39, Yunmai 42, Yunmai 47, and Fengyin 03-2, were positive for the marker. The pedigree of Yunmai 39 is *Secale cereale* L./Fuli wheat/Fticher's, and Yunmai 42 is rust-resistant 782/*Secale cereale*/Fuli wheat//YKLO-PAM"S". As *S. cereale* is susceptible to powdery mildew, *Pm16* might originate from common wheat Fuli. The genealogy of Yunmai 47 is 852-18/852-181//86-4437-75/3/822-852/785// 842-929/4/Gangu436/5/923-3763. As most of genotypes in this pedigree are breeding line numbers, it is impossible to identify the donor for the powdery

mildew resistance gene. Fengyin 03-2 originated from Anmai 5 (L9288022-2-1/Xingnong 5) in Guizhou. *Zhao et al. (2007)* found that Anmai 5 was highly resistant to powdery mildew, so *Pm16* in Fengyin 03-2 may come from Anmai 5. The pedigree of Feng1124 is E33/58769-6. E33 is an excellent powdery mildew resistant stock imported from Mexico and may contain *Pm16*.

Gene *Pm21* is derived from *Haynaldia villosa*, located on the short arm of chromosome 6V (6VS). As this gene has a wide resistance spectrum of *Bgt* isolates in the world, it has been widely studied (*Cao et al., 2011*; *Wu et al., 2019*). *Pm21* provides a high level and stable resistance in different genetic backgrounds. Meanwhile, wheat cultivars that carry this gene usually have excellent other agronomic traits. Therefore, this gene has been widely deployed in Sichuan Basin and southern Gansu since the middle 1990s. Since then, *Pm21* has been widely used in different wheat production areas in China (*Zhan et al., 2010*). *Jiang et al. (2014)* identified 7.4% of the tested 118 Chinese cultivars contained this gene using marker Scar1265 closely linked to *Pm21*. Our results showed that cultivars Kunmai 4, Yixi 2003-64, and De 4-8 contain this gene. Kunmai 4 has ALB"S"/BOW"S" in its pedigree, and ALB"S"/BOW"S" is a Chile wheat line highly resistant to powdery mildew. In addition, *Li et al. (2012)* found Kunmai 4 was highly resistant to all tested *Bgt* isolates. We found that this cultivar likely have *Pm21*. As the genealogical information is not available for Yixi 2003-64 and De 4-8, we could not identify the *Pm21* donor in these cultivars. Unfortunately only these three cultivars (4.3%) potentially have *Pm21* among the 69 tested wheat cultivars and breeding lines from Yunnan. *Pm21* should be pyramided with other effective genes to developing wheat cultivars with durable resistance to powdery mildew.

## CONCLUSIONS

Breeding resistant cultivars is the most cost-effective and eco-soundly strategy to protect wheat from disease. In the present study, the seedling resistance of 69 main wheat cultivars in Yunnan Province were evaluated using 6 isolates of *Bgt*. Overall, the seedling resistance level of wheat cultivars to powdery mildew resistance were very poor in Yunnan Province. Based on this, the presence of genes *Pm3*, *Pm8*, *Pm13*, *Pm16*, and *Pm21* in these cultivars were detected using gene specific DNA markers. Six cultivars were found likely to have *Pm3*, 18 were likely to have *Pm8*, five were likely to have *Pm16*, and three were likely to have *Pm21*. No cultivar was found to carry *Pm13*. The information on the presence of the *Pm* resistance genes in Yunnan wheat cultivars can be used in future wheat durable disease breeding programs.

## ACKNOWLEDGEMENTS

We appreciate very much to Prof. Xianming Chen from Department of Plant Pathology, Washington State University, Pullman, WA, USA for critical reading and revising of our manuscript.

## Funding
This study was supported by the National Natural Science Foundation of China (No. 31701738). The funders had no role in study design, data collection and analysis, decision to publish, or preparation of the manuscript.

## Grant Disclosures
The following grant information was disclosed by the authors:
National Natural Science Foundation of China: 31701738.

## Competing Interests
The authors declare there are no competing interests.

## Author Contributions
- Xianxin Wu conceived and designed the experiments, performed the experiments, analyzed the data, prepared figures and/or tables, and approved the final draft.
- Qiang Bian and Yuanhu Xuan performed the experiments, prepared figures and/or tables, and approved the final draft.
- Yue Gao and Xinyu Ni analyzed the data, prepared figures and/or tables, and approved the final draft.
- Yanqiu Sun analyzed the data, authored or reviewed drafts of the paper, and approved the final draft.
- Yuanyin Cao conceived and designed the experiments, authored or reviewed drafts of the paper, and approved the final draft.
- Tianya Li conceived and designed the experiments, prepared figures and/or tables, authored or reviewed drafts of the paper, and approved the final draft.

## Data Availability
The raw measurements are available in the Supplementary Files.

## Supplemental Information
Supplemental information for this article can be found online at http://dx.doi.org/10.7717/peerj.10425#supplemental-information.

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
