# Peer review of "Evaluation of resistance to powdery mildew and identification of resistance genes in wheat cultivars"

_PeerJ, doi:10.7717/peerj.10425_

## Round 0.1 · original submission · Minor Revisions

Dear authors,

In order to improve your contribution, please address the comments of the reviewers.

Reviewer 1 ·

Basic reporting

The introduction is self-explanatory.

Objectives, Material and methods, and Discussion are also clear.

Literature Review was sufficient.

The tables are clear but there is a typo error of a figure that didn't make sense.

From a breeding point of view, the introduction explained key points that any plant breeding wants to work on to develop a resistant cultivar.

Experimental design

They use the phenotyping and genotypic approach.
Both are very valid/strong approaches to identify R-genes

Validity of the findings

Relevant results.
It provides key information for breeding programs worldwide.
As long as they add a supporting figure/evidence of the Pm16 identification and correct the “typo error” of Fig 4-2, I would recommend this article for publication.

Additional comments

a) Line 148: Molecular identification of Pm13 Is there Fig 4-2? I was not able to find it in the article. Is it missing?
b) Line 153 – 158: Where is the figure/evidence for the identification of Pm16? Do you have a figure?
c) The introduction is self-explanatory. From a breeding point of view, the introduction explained key points that any plant breeding wants to work on to develop a resistant cultivar. Objectives are also clear.
d) Well written discussion. I enjoy the idea of pyramiding genes in elite wheat lines.
e) Very nice review of the literature.

·

Basic reporting

This manuscript evaluate seedling resistance of 69 main wheat cultivars from Yunnan of China, and also conformed the Pm genes in these cultivars. The information can be used in future wheat disease breeding. However, several revisions should be carried out based on the suggestions listed blew. After revisions, the manuscript can be accepted for pubilcation in PeerJ

Experimental design

No comment

Validity of the findings

No comment

Additional comments

1.Up to now, 61 loci (Pm1-Pm66) have been characterized to provide powdery mildew resistance, but not 58, please update the progress.
2.The documented Pm stocks in Table 2 were not near-isogenic lines, but in different genetic backgrounds.
3.Please describe the detail method to avoid cross-infection of different isolates after inoculation.
4.Different IT scores have detailed indicators, please re-describe them.
5.All the identification of the Pm genes were based on the amplification of parents but not populations of resistance and susceptible parents, so you can not confirm one cultivar certainly has one related Pm gene only by the special band comparison between the tested cultivars and Pm gene stocks. You need use “is most likely” to explain one cultivar is most likely have one special Pm gene based on the special band.
6.Yunnan is not the main wheat growing regions in China.
Why do you selected Pm3, Pm13 and Pm16? These three genes rarely exist in Chinese cultivars, especially there are no commercial varieties to be reported to have Pm13 and Pm16. Chinese cultivars mainly have Pm2, Pm4, Pm8 and Pm21.

---

## Round 0.2 · accepted · Accept

Dear authors, I have checked that the corrections have been made as indicated by the reviewers, so I congratulate you for your contribution to such a prestigious journal.

The Section Editor made the following comments that you might consider in the proofing stage.

> In general, SSR mapping manuscripts are usually treated as simple reports which require more just cause for publication. However, the authors do make a point that there is a need to develop resistant germplasm to the pathogen and do a fine job in providing some background for the resistance genes that they are characterizing and its needed importance for agriculture.
> It would be great to have images of example infection types to better illustrate the scoring system, but the wide range of differentials tested may provide a good idea of the gene distribution. Otherwise, I feel this is a successful manuscript, but I do see a few minor edits which are needed and they are listed below.
> EDITS LINE NO: / BEFORE / AFTER / [COMMENTS] LINE 16: / alos / also / [.] LINE 58: / translation / translocation / [.] LINE 86: / cultivars use / cultivars using / [.] LINE 104: / suing / using / [.] LINE 108: / with glass / with a glass / [.] LINE 110: / after planting). / after planting) . . . . / [incomplete sentence.] LINE 130: / Five wheat cultivar, / Five wheat cultivars, / [.] LINE 135: / contains a more / contains more / [.] LINE 167: / Discussion / . / [place ‘Discussion’ on new line.] LINE 171: / stages of history. / stages in history. / [.] LINE 191: / been play an / been playing an / [.] LINE 194: / than wheat cultivars grown in 50% of the / 50% of the wheat cultivars grown in / [.] LINE 253: / seedling resistant / seedling resistance / [.] LINE 254: / Base on it, / Based on this, / [.] LINE Table 4: / Percentagess / Percentages / [.] LINE Table 4: / . / . / [are there missing annotations for footnotes 1-4?]